# Analysis of the Current Situation of Big Data MOOCs in the Intelligent Era Based on the Perspective of Improving the Mental Health of College Students

Hongfeng Sang [1], Liyi Ma [2,*] and Nan Ma [1]

1 Beijing Institute of Artificial Intelligence, Beijing University of Technology, Beijing 100124, China; sanghongfeng@bjut.edu.cn (H.S.); manan123@bjut.edu.cn (N.M.)
2 School of Management, Beijing Union University, Beijing 100101, China
* Correspondence: gltliyi@buu.edu.cn

**Abstract:** A three-dimensional MOOC analysis framework was developed, focusing on platform design, organizational mechanisms, and course construction. This framework aims to investigate the current situation of big data MOOCs in the intelligent era, particularly from the perspective of improving the mental health of college students; moreover, the framework summarizes the construction experience and areas for improvement. The construction of 525 big data courses on 16 MOOC platforms is compared and analyzed from three aspects: the platform (including platform construction, resource quantity, and resource quality), organizational mechanism (including the course opening unit, teacher team, and learning norms), and course construction (including course objectives, teaching design, course content, teaching organization, implementation, teaching management, and evaluation). Drawing from the successful practices of international big data MOOCs and excellent Chinese big data MOOCs, and considering the requirements of authoritative government documents, such as the no. 8 document (J.G. [2019]), no. 3 document (J.G. [2015]), no. 1 document (J.G. [2022]), as well as the "Educational Information Technology Standard CELTS-22—Online Course Evaluation Standard", recommendations about the platform, organizational mechanism, and course construction are provided for the future development of big data MOOCs in China.

**Keywords:** big data; intelligent era; MOOC construction; investigation; analysis

## 1. Introduction

The advent of "big data" represents a new technological trend in the internet era, ushering in novel opportunities for societal development. The World Economic Forum's *Future of Jobs Report for 2020* highlights that, in the wake of the global spread of the pandemic, professions with high information-technology components are evolving rapidly, with priorities for business leaders still centered around the utilization of cloud computing, big data, and e-commerce. The pervasive diffusion and application of big data throughout various industries is furnishing the economy with fresh resources and new impetus. Big data has laid a robust foundation for the development of the intelligent era, which requires intellectual tasks to be subjected to smart processing—tasks that can be facilitated by data analysis. The advent of the intelligent era has had a comprehensive impact on traditional education through the ascendance of artificial intelligence [1]. Intelligence is rooted in education [2], and the value of higher education in the intelligent era will be further enhanced by the promotion of artificial intelligence, which drives educational reform [1] and the emergence of more interdisciplinary studies in universities, becoming the main source of scientific and technological innovation [3]. While the world is promoting artificial intelligence education, the importance of data literacy cannot be overstated. In 2011, Carlson J. R. et al. of Purdue University proposed an eight-core competency framework [4] for data literacy, encompassing data awareness, data knowledge and skills, and data ethics

and standards [5]. In September 2015, the State Council of China issued an *Action Plan for Promoting the Development of Big Data*, acknowledging data as strategic resources of the era, and emphasizing the cultivation of data talents. Big data has brought about significant changes in the patterns of scientific and industrial innovation across various nations, attracting high levels of attention in academia and industry alike. Thus, there is an urgent need to accelerate the training of big data talents, with the aim of satisfying personnel demands in big data and artificial intelligence-based industries.

Since 2012, the online education market has rapidly developed worldwide. Massive open online courses (MOOCs) have greatly expanded teaching hours and spaces, reduced education costs, and broadened the reach of high-quality educational resources. With strong learning interactivity and timeliness, MOOCs have become important modalities in higher education. The Ministry of Education has successively issued documents, such as the no. 3 document (J.G. [2015]) [6] and no. 1 document (J.G. [2022]) [7], requiring the strengthening of the construction and application management of open online courses in higher education institutions. With the rapid development of data science and the big data industry, MOOCs related to big data have gradually increased. Universities, research institutions, social organizations, and even companies have released big data-related MOOCs on domestic and foreign MOOC platforms, such as Coursera, edX, and China University's MOOC. The establishment of data science subjects has also promoted changes in related teaching content, methods, modes, and teaching management systems and mechanisms, bringing new opportunities and challenges to the teaching reform and development of big data courses. Scholars have begun to pay attention to the development of big data courses in the new era. This includes the following aspects:

(1) Research on big data course systems, including a phased introduction of core courses, like distributed systems, cloud computing, data acquisition, data processing, big data mining and analysis, big data prediction and artificial intelligence, data visualization, and other core courses for application-oriented big data talent training course groups in the context of new engineering disciplines. Scholars also believe that, in addition to some basic courses of the major-related courses (such as database principles, computer networks, etc.), the knowledge and ability requirements in the course settings and training objectives are not fully matched, and the students' numeracy skills are lacking [8] for the training of big data management and application majors in regard to new liberal arts disciplines [9].

(2) An investigation into the MOOCs of specific big data majors. Scholars have analyzed and provided development suggestions for 57 data visualization MOOCs from the aspects of teaching content, course difficulty, prerequisite knowledge, teaching staff, assessments, and certification [10]. They also conducted investigations and analyses into the data analysis courses of Coursera and edX platforms [11].

(3) Research into the learning behaviors and effects of big data MOOCs, including modeling users' learning behaviors, analyzing the relationship between learning behaviors and final results, predicting subsequent learning behaviors, and so on [12]. Based on the implementation method before, during, and after MOOCs, scholars explore how to ensure the effectiveness of implementation, thereby improving students' proactive learning capabilities, learning interests, enthusiasm, and problem-solving abilities [13].

The above research presented a survey and research into the big data course system, professional course offerings, and user learning behaviors in the new era, and promoted the reform of relevant teaching content, methods, and systems, to a certain extent. More researchers have mainly focused on online evaluations, personalized recommendations, and the cloud platform construction of MOOCs by using big data technology [14–16]. However, few scholars have conducted general surveys and comparative analyses on the opening of big data MOOCs in China and abroad. Based on 525 big data MOOCs from 16 major MOOC platforms at home and abroad, this paper constructs a three-dimensional MOOC analysis framework, encompassing a *platform, organizational mechanism,* and course construction, revealing the common points and differences of relevant research at home

and abroad in this field, and providing references for the research and development of big data MOOCs in China.

## 2. Research Framework and Target

### 2.1. Research Framework

In February 2022, the Chinese Ministry of Education issued the no. 1 document (J.G. [2022]) on MOOCs; the objective was to standardize the teaching management of open online courses and maintain the teaching order of open online courses, which explicitly defines six aspects: teaching subjects, teaching norms, learning norms, platform guarantees, platform supervision, and collaborative mechanisms. In particular, "teaching subjects" are required to manage MOOCs and prioritize higher educational institutions as the main teaching subjects. Regarding "teaching norms", higher educational institutions are obligated to strengthen their management over MOOC instructors, while "learning norms" require the rigorous implementation of student norms for online learning and exam disciplines. Furthermore, "platform guarantees" demand the implementation of a self-supervision mechanism for MOOCs, while "platform supervision" entails established regulations for online course supervision. Finally, the "collaborative mechanism" necessitates the establishment of interdepartmental cooperation. The above contents are mainly related to the MOOC platform construction and the organizational mechanism of universities. Among them, the successful implementation of MOOCs lies in the platform's operation, supervision, and governance, as well as in the three-in-one mechanism established by higher educational institutions, instructors, and students.

In 2019, the Chinese Ministry of Education issued the no. 8 document (J.G. [2019]) on MOOCs, the goal of which was to form a first-class undergraduate curriculum system with Chinese characteristics at a world-class level, and to build a higher talent training system [17]. Some suggestions have been put forward for the construction of first-class undergraduate courses, such as innovative ideas, goal orientation, enhancing teacher capabilities, innovating teaching methods, improving student evaluations, and stimulating teaching and strengthening management. According to the document, this paper argues that reasonable course construction is the core of the smooth implementation of MOOCs.

On 7 June 2002, China's Education Technology Standards Committee officially issued the "Educational Information Technology Standard CELTS-22—Online Course Evaluation Standard" (draft standard for comment), referred to as the CELTS-22 standard, which mainly studies the evaluation of online courses from four dimensions: course content, instructional design, interface design, and technology.

We synthesized the content of platform construction, organizational mechanism construction, and course construction required by the above three documents; we constructed a three-dimensional MOOC analysis framework for the platform-organizational mechanism and course construction, as shown in Figure 1.

### 2.2. Research Target and Method

The current study employed a combination of online surveys and content analysis methods to investigate MOOC platforms, including major domestic and foreign MOOC platforms and the "National Higher Education Wisdom Education Platform" (NHWEP). NHWEP is a comprehensive online open course platform developed and operated by the Higher Education Press under the commission of the Chinese Ministry of Education. The primary goal of NHWEP is to gather the best courses developed by the best universities and instructors both domestically and abroad, aiming to become the world's largest and most comprehensive national higher education wisdom education platform, covering the widest range of subjects and attracting the most users. The initial 20,000 online courses were selected from over 50,000 high-quality courses, developed by 1800 universities, covering 92 majors in 13 disciplines. Sixteen MOOC platforms, both domestic and foreign, were selected as the research targets based on the navigational bar of big data courses that could be searched on the websites. First of all, using "big data" as the keyword, the crawler

software is applied to login to the MOOC platform for course retrieval, retrieve the courses related to big data, and crawl the relevant course attribute information, including (but not limited to) the following fields: first-level classification, second-level classification, course name, course starting unit, country, difficulty level, class hours, credits, platform name, instructor, faculty team, number of participants, course introduction, detailed introduction, course objectives, course outline, course design, certification, prior knowledge, course characteristics, teaching language, assessment criteria, etc. (Different platforms encompass distinct fields; the above information is a summary of the main fields of each platform). Secondly, mathematical statistical analysis and text content analysis are carried out on the curriculum information that has been crawled. Among them, Coursera, edX, and FutureLearn are the main foreign platforms. The main domestic platform choice is the "National Higher Education Smart Education Platform" included in the MOOC platform. As of January 2022, a total of 525 big data-related courses were collected from these platforms, including 325 and 200 courses from foreign and domestic platforms, respectively. Detailed statistics are presented in Table 1.

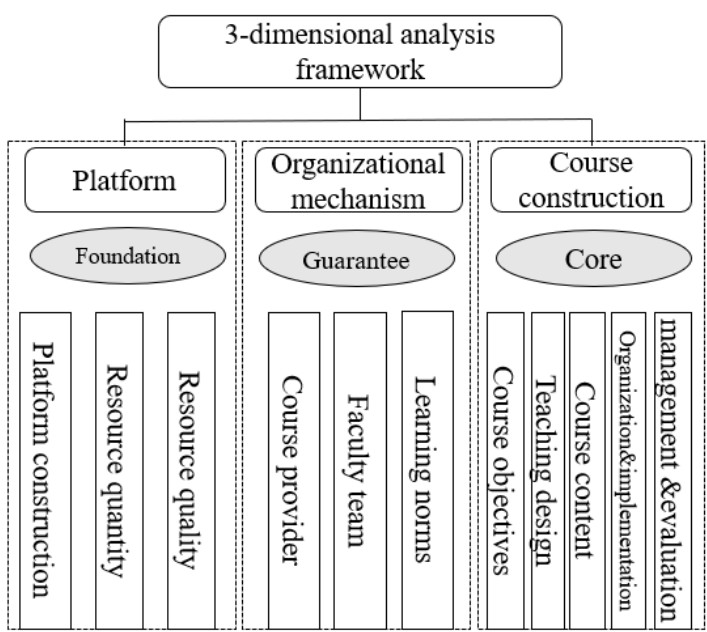

**Figure 1.** Three-dimensional MOOC analysis framework.

**Table 1.** Statistics of the number of big data-related courses offered on MOOC platforms.

| Foreign MOOC Platforms | Number of Courses | Domestic MOOC Platforms | Number of Courses |
|---|---|---|---|
| Coursera | 181 | Zhihuishu | 58 |
| edX | 123 | Chinese University MOOC | 50 |
| FutureLearn | 20 | XuetangX | 49 |
| Udacity | 1 | Xueyin Online | 18 |
| | | Alibaba Cloud Classroom | 12 |
| | | Zhengbao Cloud Classroom | 3 |
| | | Chinese MOOCs | 3 |
| | | Chongqing Online Open Course Platform for Higher Education | 2 |
| | | Rongyou Xuetang | 2 |
| | | Zhejiang Online Open Course Sharing Platform for Higher Education | 1 |
| | | CNMOOC | 1 |
| | | Gaoxiaobang | 1 |
| Total | 325 | Total | 200 |

Source: collected by the author.

## 3. Analysis of Big Data Courses on MOOC Platforms

### 3.1. Platform Dimension

#### 3.1.1. Platform Construction

According to the no. 1 document (J.G. [2022]) promulgated by the Chinese Ministry of Education, the platform construction is primarily reflected in self-regulatory mechanisms, regulatory systems, and collaborative mechanisms. Platforms offering credit courses must strictly comply with network security level protection protocols, reinforce the monitoring of learning processes, and execute the systematic big data monitoring of open online course teaching procedures. Regarding privacy policies, both domestic and foreign platforms have pledged to safeguard the integrity of users' personal information. They have formulated relevant policies related to the collection and usage of personal information, the utilization of cookies or similar technologies, and the management, sharing, transfer, and disclosure of such information, in addition to stipulations on the retention, storage, and overall protection of personal information security. Furthermore, to ensure learning integrity, the UOOC Alliance Course issued a series of measures in 2019. As such, once a case of cheating is discovered, the student's learning account is frozen immediately for a year. If the school detects two or more instances of academic dishonesty, the student's learning account is permanently frozen or deleted, and the student's school is required to impose disciplinary action based on relevant academic and disciplinary regulations. Additionally, with respect to platform operations, the Chinese University MOOC offers interactive features with social media giants, including WeChat, QQ, and Weibo. By harnessing these resources, students can share courses on these platforms and receive course progress reminders through WeChat. XuetangX provides an intelligent teaching assistant called Xiaomu, which offers self-service utilities, such as explaining knowledge concepts, searching course materials, answering common platform questions, and recommending academic resources.

#### 3.1.2. Resource Quantity

Based on the findings of a recent study, 89 universities or institutions from 15 different countries currently offer 325 MOOCs related to big data on foreign MOOC platforms in 10 languages, namely, English, German, Russian, Korean, Portuguese, Spanish, Italian, French, Japanese, and Chinese. The United States has the highest number of big data MOOCs offered (with a total of 202 courses), followed by China (with 35). Other countries with more than 5 big data MOOC offerings include Russia (15), the United Kingdom (11), Mexico and Spain (8 each), and the Netherlands (7). In terms of the subjects offered, computer science has the highest number of courses (260), followed by business management (23), mathematics and statistics (10), biology, medicine, and health (6 each), environment and earth (4), and education and ethics (3 each). History only has 2 courses, while electronics, law, economics, architecture, finance, physics, psychology, and social sciences have 1 course each. The level of difficulty of big data MOOCs offered by foreign platforms varies, as indicated in Table 2.

**Table 2.** Difficulty statistics of foreign MOOC platforms.

| Level | Number |
|:---:|:---:|
| Beginner | 137 |
| Intermediate | 96 |
| Advanced | 32 |
| Professional | 5 |
| Mixed | 55 |
| Total | 325 |

Source: collected by the author.

On domestic MOOC platforms, 85 universities or institutions offer 200 big data MOOCs. From the perspective of first-level disciplines, engineering has the highest number (141 courses), followed by management (37), science (10), and economics (7). Other

disciplines include law, education, medicine, and literature. Similar to the level of difficulty on foreign MOOC platforms, more than half of big data MOOCs provided by domestic platforms are beginner or introductory courses.

### 3.1.3. Resource Quality

In terms of platform interoperability, many MOOC platforms are actively breaking down language barriers in education. For example, the foreign platform "Coursera" provides English, Chinese, Japanese, and Spanish subtitles for course-related videos, while "Udacity" has a Chinese interface, allowing users to search for courses in Chinese [18]. The major domestic MOOC integration platform, NHWEP, offers courses in multiple languages, linking the two international platforms "iCourse" and "XuetangX", catering to students from different parts of the world. "CNMOOC" has an English interface, but the course resources are mainly in Chinese. In terms of original course construction, Chinese MOOC platforms face a shortage of original resources [17]. Currently, there are only 11 big data courses with the National Quality Course title, accounting for only 5.5% of the total number of big data MOOCs in China. These courses are spread across four MOOC platforms: Chinese University MOOC, XuetangX, Rongyou Xuetang, and Gaoxiaobang. They only account for 33.3% of the platforms with big data MOOCs in China, and 25% of higher education MOOC platforms (which integrate 20 MOOC platforms). Among these 11 courses, all the speakers have senior titles, with 64% being professors and 36% being associate professors. Apart from two courses, the other nine courses are offered by double first-class universities. The lecturers of these courses have strong scientific research abilities, such as Professor Wang Guoyin, the team leader of "Big Data Analysis and Processing", who is a Yangtze River Scholar–specially appointed professor in the field of computer science and technology and a national candidate for the "Millions of Talents Project for the New Century". Professor Wang Jianmin, the team leader of "Fundamentals of Big Data Systems", has been awarded the National Outstanding Youth Science Fund and won the second prize of the National Science and Technology Progress Award. He is the chief expert group leader for the national 863 Program and has been recognized as a New Century Excellent Talent and a leading figure among young and middle-aged science and technology professionals by the Ministry of Science and Technology. Many course lecturers have won provincial and ministerial-level natural science awards and their research achievements in the field of teaching and research are significant, as shown in Table 3.

### *3.2. Organization Dimension*
### 3.2.1. Course Provider

According to the 325 big data MOOCs offered on foreign platforms, the majority of course providers are universities. In addition, 81 courses are offered by companies and foundations, such as Microsoft, IBM, Google, Cisco, PwC, NetEase, Linux Foundation, and the International Monetary Fund. There is also one bank (Inter-American Development Bank) that offers five introductory-level courses. Overall, the introductory-level courses mainly focus on the applications of big data in various fields, providing students with entry-level courses. For example, the courses offered by the Inter-American Development Bank are related to data in agriculture policy and public policy effectiveness. Similarly, the big data MOOCs offered on domestic platforms primarily come from universities, with businesses such as Microsoft Asia Research Institute, IBM, Intel Asia-Pacific Research and Development, Ltd., Huawei, and Alibaba Cloud offering supplementary courses.

**Table 3.** Big Data MOOCs that have won the "National Quality Course" award.

| Platforms | Year | Course Title | Opened by | First-Rate Universities and Disciplines or Not? | Main Creators and Those Who Are Primarily Responsible for the Course | Title of the Lecturer | Collaborating Institutions | Number of Times the Course Has Been Offered | The Number of Participants Who Have Taken the Course |
|---|---|---|---|---|---|---|---|---|---|
| Chinese University MOOC | 2014 | Big Data Algorithm | Harbin Institute of Technology | Yes | Wang Hongzhi | Professor | / | 5 times, providing self-study mode | 207,432 |
| | 2016 | Biological Big Data | Fujian Agriculture and Forestry University | No | He Huaqin | Professor | Chinese Academy of Sciences and enterprises | 11 times | 35,667 |
| | 2016 | Playing with Data in Python | Nanjing University | Yes | Zhang Li | Associate professor | / | 13 times | 405,270 |
| | 2017 | Python Data Analysis and Presentation | Beijing Institute of Technology | Yes | Song Tian | Professor | / | 12 times | 492,528 |
| | 2017 | The Principle and Application of Big Data Technology | Xiamen University | Yes | Lin Ziyu | Associate professor | / | 10 times | 235,803 |
| | 2017 | Analysis and Application of Business Data | Jiangsu Vocational Institute of Commerce | No | Wu Honggui | Professor | / | 11 times | 38,156 |
| XuetangX | 2019 | Fundamentals of Big Data Systems | Tsinghua University | Yes | Wang Jianmin | Professor | / | 6 times | 94,818 |
| | 2019 | Advanced Big Data System | Tsinghua University | Yes | Wang Zhi | Associate professor | / | 6 times | 34,393 |
| | 2019 | Big Data Machine Learning | Tsinghua University | Yes | Yuan Chun | Associate research fellow | / | 7 times | 45,283 |
| | 2019 | Big Data Machine Learning | Tsinghua University | Yes | Wu Yongwei | Professor | Alibaba Cloud | Self-study mode | 71459 |
| Rongyou Xuetang | 2017 | An Introduction to Data Science | Renmin University of China | Yes | Chao Lemen | Associate professor | / | 8 times | 34,138 |
| Gaoxiao-bang | 2019 | Big Data Analysis and Processing | Chongqing University of Posts and Telecommunications | No | Wang Guoyin | Professor | / | 7 times | 3038 |

Source: collected by the author.

### 3.2.2. Faculty Team

High-level faculty teams are important for promoting teaching content reforms, developing high-quality teaching resources, and improving teaching quality [19]. Based on the analysis of data from big data course teams at home and abroad, faculty teams are mainly composed of individual lecturers, probably due to the high dependence of big data courses on the lecturer's professional background, which mainly determines the course content [20]. Team cooperation mainly consists of 3–5 people, up to 10 people (such as the team led by Prof. Zhong Xueling in the "Application of Big Data Tools" course at Guangdong University of Finance). Among them, there are many teachers with associate professor titles (and above), with senior teachers being the main lecturers. The number of part-time instructors from enterprises is relatively small, and courses developed through industry–university–research collaboration are few. For example, among the 11 national quality courses, only 2 courses have research institutes and enterprises involved in co-construction. In addition, for MOOCs launched by companies such as IBM, Huawei, Alibaba Cloud, and Microsoft, most of the lecturers are experienced vice presidents, senior developers, and data analysts who work closely with practical engineering.

### 3.2.3. Learning Norms

According to the requirements of the Ministry of Education's no. 1 document (J.G. [2022]), university students are expected to comply with their respective school's course selection requirements by taking online open courses through the academic system, signing online learning honesty commitments, abiding by course learning disciplines and exam disciplines, and strictly prohibiting the lending of accounts and course-brushing behaviors. Any violations or infractions found will result in relevant measures being taken by the relevant authorities and legal responsibilities being pursued in accordance with the law. Course-brushing behavior is highly covert, and according to public reports, in 2020, the Chaoyang City Public Security Bureau in Liaoning Province cracked the first nationwide case of "pay-per-view course brushing", which involved over 40 mainstream MOOC platforms. From 2019 to 2020 alone, over 7.9 million students nationwide purchased "course-brushing" services, with a "course-brushing" volume exceeding 79 million. These students used illegal software to brush courses during the learning process, severely disrupting MOOC teaching activities. As of April 2022, a search using the keyword "course-brushing" on Baidu's website still yields a small number of posts on certain forums, with "cracking" being the main topic, and comments on "how to brush courses on online platforms" or "how to brush courses quickly".

### *3.3. Course Structure Dimension*

### 3.3.1. Course Objectives

The objectives of big data MOOCs in China differ, depending on the characteristics of individual schools or programs, but based on research findings, only courses available on the Zhihuishu platform provide a course design module that offers information on course backgrounds, objectives, and design principles. In general, goals for big data MOOCs in China lack clear definition, with many courses failing to explicitly articulate learning goals. Furthermore, such courses lack the comprehensive design and interpretation of these learning goals, which should focus on students acquiring knowledge, skill development, and value formation.

### 3.3.2. Teaching Design

The design and implementation of online teaching must be tailored to learning objectives, with a focus on interaction and feedback, cultivating a sense of student autonomy, and providing convenient online resources. In our research on big data MOOCs, we found that the main learning resources include micro-video lectures, audio, images, PPT courseware, e-books, and text. Micro-video lectures are the most common resource, followed by PPT courseware, e-books, and text. Some courses, such as "Big Data Machine Learning" by Professor Yuan Chun at Tsinghua University, provide chapter knowledge graphs to help

students grasp the overall content of the chapter, and offer supplementary materials outside the curriculum to broaden their perspectives. Additionally, Associate Professor Ding Zhaoyun's "Data Mining" course at the National University of Defense Technology uses information technology to achieve the dynamic combination of teacher lecturing scenes, PPT, animations, and videos, while adopting a video interaction mode to improve visual effects and learning experience. Overall, big data MOOCs rely heavily on lecture-based delivery. This one-dimensional approach lacks methodological innovation, thereby limiting teaching effectiveness and learning quality. In terms of course evaluation design, big data MOOCs rely primarily on teacher evaluations, with automated evaluations based on the MOOC platform serving as secondary measures. The most common evaluation methods are "unit tests + exams", followed by "unit tests + exams + course discussions".

### 3.3.3. Course Content

After analyzing and categorizing 525 domestic and overseas big data courses, it was discovered that MOOCs on big data mainly cover seven areas: data acquisition and processing, data analysis and mining, data visualization, big data management, cloud computing and distributed platforms, artificial intelligence, and big data applications. These findings are detailed in Table 4.

The data presented in Table 4 demonstrate that MOOCs dedicated to big data exhibit distinctive characteristics, which underscore their construction. These courses are generally characterized by the use of popular programming languages, such as R and Python, the presentation of abundant examples during instructions, and the multi-disciplinary application of big data in teaching, resulting in comprehensive coverage of the entire data science pipeline. For instance, Associate Professor Li Hui of the School of Software at Shandong University presents "Non-Relational Database Technologies", utilizing practical development projects as case studies, introducing the application of the NoSQL database MongoDB in the development of e-commerce websites. Similarly, the "Big Data and Finance" course offered at Central University of Finance and Economics by Associate Professor Peng Yuchao is oriented toward employing big data analytical methods to address finance-related issues while comprehensively exploring the applications of big data in traditional finance, internet finance, credit ratings, and regulatory technology. This approach provides a platform to increase students' understanding of cutting-edge finance-related developments.

### 3.3.4. Teaching Organization and Implementation

Several courses have introduced the latest scientific research outcomes in the field of big data into classroom teaching. They focus on practical technical skills and integrate classic case studies into the course curriculum to improve the effectiveness of the course. For instance, Professor Ren Zongwei of Harbin University of Commerce delivers the course "Big Data and Artificial Intelligence" based on the latest industry development requirements. The course introduces cutting-edge theories and methods, strengthens the internal links between big data and artificial intelligence, and designs course content in the form of theory, practical methods, and case studies. Professor He Huaqin of Fujian Agriculture and Forestry University created the course "Biological Big Data" in response to the development of biological genome sequencing technology, focusing on the biological big data obtained from next-generation sequencing technology. There are no similar courses offered domestically or abroad. This course helps students understand the characteristics, values, and applications of biological big data and enables them to master analytical skills to process the data. In foreign big data MOOCs, the course material is designed for learners at various levels, ranging from introductory to intermediate, advanced, and mixed levels, to facilitate differentiated teaching practices. However, the domestic big data MOOCs fall short of paying attention to individual differences among students. The implementation of online, offline, in-class, and extracurricular interactive teaching management is crucial in promoting personalized self-learning among students.

**Table 4.** Distribution of teaching content.

| Teaching Content | Course Title | Proportion |
|---|---|---|
| Data acquisition and processing | Acquiring and Analyzing News and Academic Data Related to the COVID-19 Pandemic; Big Data Collection and Storage; Geographic Data Processing and Charting Techniques; Excel Data Processing and Analysis; Introduction to Big Data: Mathematical Foundations and Applications; Big Data Processing and Analysis; GNSS Measurement and Data Processing; Finance Data Processing Technology; Data-Driven Decision Making: Market Research Practice; Data Journalism; Discovering the Beauty of Data with SPSS; Introduction to Big Data Analytics and Applications; Big Data: Processing and Analysis; Data Storage and Processing; Statistics and Data Science; Statistical Analysis with R Language. | 11% |
| Data analysis and mining | R Language Data Analysis; Python Data Analysis Practice; Python Foundation for Big Data Analysis; Python Data Analysis Practice; Analysis of Business Statistics; Big Data Analysis and Forecast Technology; Python Data Analysis and Application: R language Data Analysis and Mining; Big Data Analysis and Visualization; Python Language and Economic Big Data Analysis; Introduction to Data Analysis and Processing- Backup; Excel Data Processing and Analysis; Data Analysis Using Python; R Language Programming (Chinese version); SPSS Multivariate Analysis; Time and Space Big Data Analysis and Mining Actual Combat; Introduction to Big Data; Data Warehouse and Data Mining; Python Data Mining; Application of Big Data Tools; Data Mining Techniques for Big Data Analysis: Introduction to Data Mining; Data Mining. | 23% |
| Data visualization | Microsoft Big Data Visualization; Python Data Analysis and Data Visualization; Data Visualization with Python; Insight into Data: Introduction to Data Analysis and Visualization; Data Analysis: Visualization and Dashboard Design; Microsoft Excel—Data Visualization, Excel Charts and Graphics; Big Data: Data Visualization; Visualizing Data with Python; Understanding and Visualizing Data with Python; Data Visualization with Tableau; Data Science: Visualization; Data Visualization and Storytelling; Information Visualization; Data Visualization—Analysis and Design. | 13% |
| Big Data Management | Using HBase for Real-time Management of Your Big Data; How to Move Data to Hadoop; How to Access Data on Hadoop with Hive; Building Data Warehouse for Business Intelligence; Excel: Data Management; Big Data: Modeling and Management System; Managing Big Data with MySQL; Essentials of Database Management; Data Management, Security, and Robotic Operating System as a General Tool in IoT; Managing Big Data with R Language and Hadoop; Agile Data Science for Product Management; Non-Relational Database Technologies; Advanced Big Data Systems; Advanced Big Data Systems. | 13% |
| Cloud computing and distributed platforms | Big Data Platform Core Technologies; Big Data Platform Technologies; Principles and Applications of Big Data Technology; TensorFlow Machine Learning Based on Google Cloud Platform; Introduction to Big Data and Cloud Computing; Cloud Computing and Big Data Technology; TensorFlow: Data and Deployment; Managing Big Data in Cluster and Cloud Storage; Advanced Machine Learning with TensorFlow on Google Cloud Platform; Data Engineering, Big Data, and Machine Learning on Google Cloud Platform; Cloud-based Delivery of Data Warehouse; Cloud Computing. | 16% |
| Artificial intelligence | Automatically Calibrating Intersection Topology Information with Trajectory Data; From Data to Decision: Three Machine Learning Tasks, Three Structural Transformations; Big Data and Artificial Intelligence; Data Intelligence and Applications; IBM Artificial Intelligence Application Professional Certificate; Big Data, Artificial Intelligence, and Ethics; Workflow of Artificial Intelligence: Data Analysis and Hypothesis Testing; Data Ethics, Artificial Intelligence, and Responsible Innovation; Machine Learning Engineer; Master's Program in Machine Learning and Data Science; Machine Learning Based on Big Data; Reinforcement Learning; Deep Learning Engineer; Data Science: Machine Learning; Data Science: Statistics and Machine Learning; Fundamentals of Data Science: Prediction and Machine Learning; Vertex Projects in Data Science and Machine Learning; Machine Learning with Data Science and Analytics; Applications of Mathematics in Machine Learning. | 16% |
| Big data application | Agri-Big Data; Big Data Marketing and Management in Chain Enterprises; Knowledge Management and Big Data in Business; Machine Learning and Reinforcement Learning in Finance; Forestry Big Data and Artificial Intelligence; Analyzing Box Office Data with Ploly and Python; Analyzing Box Office Data with Seborn and Python; Big Data and Urban Planning; Big Data for Smart Grids; Knowledge Epidemic Map—Application of AI and Big Data in Intelligent Services for COVID-19; Acquisition and Analysis of News and Academic Data for COVID-19; Practical Application of Big Data at Tsinghua University: Rapid Construction of Data-driven Applications. | 8% |

Source: collected by the author.

### 3.3.5. Course Management and Evaluation

Regarding teaching management and evaluation, the main evaluation methods adopted in big data MOOCs include pre-course diagnostic evaluation (such as pre-course tests, questionnaires), in-course formative evaluation (such as embedded problems, chapter quizzes, interactive discussions), and post-course summative evaluation (such as final exams, research reports). Chapter quizzes and final exams are the most commonly used evaluation methods in big data MOOCs, followed by interactive discussions. Regarding the platform, China University MOOC and XuetangX mainly adopt evaluation methods, such as chapter quizzes, interactive discussions, research reports, and final exams. Meanwhile, Zhihuishu and Xueyin Online mainly adopt evaluation methods, such as video browsing, interactive

discussion, chapter quizzes, in-class interactions (check-ins), and final exams. Coursera's evaluation methods include quizzes, peer-graded assignments, programming assignments, etc. edX's evaluation methods include unit assignments, student peer assessments, video exams, final exams, and graduation projects.

## 4. Suggestions

In China, MOOCs were introduced in 2013, and after 8 years of development, more than 34,000 courses have been launched, with over 540 million learners and 150 million in-school learners receiving MOOC credits. The Chinese MOOC industry holds the world's top position in terms of the number of courses offered and the scale of learners. However, the construction of MOOCs focusing on big data began relatively late in China and still faces various challenges. Drawing upon successful experiences of international big data MOOCs and excellent Chinese big data MOOCs, and considering the requirements of relevant documents, such as the no. 8 document (J.G. [2019]), the no. 3 document (J.G. [2015]), the no. 1 document (J.G. [2022]), as well as the "Educational Information Technology Standard CELTS-22—Online Course Evaluation Standard", the following recommendations are provided for the future development of big data MOOCs in China.

### 4.1. Platform

Regarding the platform, it is essential to strengthen the implementation of artificial intelligence and big data monitoring to better monitor the learning process of students, detect and diagnose abnormal learning behaviors, and enhance the monitoring of the learning process. But the implementation of an MOOC platform for AI and big data monitoring may involve the collection and analysis of large amounts of students' personal data during the monitoring of their learning process, thus introducing potential risks related to data privacy and autonomy. Therefore, the MOOC platform must promise to protect the security of users' personal information and privacy, and develop sound privacy policies on how to collect and use users' personal information, how to use cookies or similar technologies, how to manage, share, transfer, and disclose personal information, and how to retain, store, and protect the security of personal information and the protection of minors, in order to prevent potential misuse of data.

Additionally, it is recommended that the platform provides more intelligent teaching and interactive functions, such as knowledge maps, difficulty gradings, grade rankings, and remedial strategies. These enhancements can serve as smart assistants for both teachers and students, augmenting the appeal of online learning.

In terms of the construction of platform course resources, although the number of big data MOOCs on domestic platforms has been increasing each year, there is still a significant gap with foreign platforms. There are many course duplications on different platforms and relatively scarce original course resources; only 11 big data courses have won the National Excellent Course title, accounting for only 5.5% of the total number of domestic big data MOOCs. Moreover, practical big data courses are also in short supply, which is not conducive to the overall development and construction of data science in China. Therefore, it is necessary to strengthen collaboration between platforms, prestigious universities, top majors, outstanding companies, research institutions, and educational institutions, to increase the number of high-quality resources, and expand the variety of big data courses. For example, the engineering micro-degree project launched by the school online adopts the learning mode taught by the university theory, enterprise application, and industry-famous teachers, and comprehensively improves the path and method of engineering education teaching quality. After students pass the assessment and complete the study of the micro-degree program, they can also obtain the certification jointly issued by the online school and collaborating enterprises and universities, leading to direct promotion opportunities with prime job enterprises. It is essential to advance theoretical foundations, basic theories, and domain applications. By developing online practice platforms based on the cloud, striking a balance between theoretical and practical-oriented courses, constructing a comprehensive

big data knowledge system, and enriching the big data MOOC learning platform, we can further the growth of big data and data science disciplines in China.

*4.2. Institutional Framework*

In both China and abroad, universities are the main institutions offering big data courses, while businesses and institutions (including research and educational institutions) play supporting roles. However, the proportion of domestic businesses and institutions offering big data MOOCs is lower than that of foreign institutions. Most universities, businesses, and institutions offer courses independently. In the early stage of the development of online education in our country, there was a one-sided view that online education was a digital version of traditional educational resources, with emphasis on technical attributes, and technology at the center. This is the main reason for the lack of collaboration between industry, universities, and research in course development. To promote the development of big data MOOCs, production, research, and education institutions should encourage collaboration, while universities with industry development characteristics should offer distinctive big data MOOCs. The combination of cutting-edge research, industry enterprise engineering, and production practice should be used to create high-quality resources and launch courses that meet the urgent demands of society across various industries, such as medical, business, finance, manufacturing, aerospace, geology, biology, etc.

Concerning faculty team building, student development should be at the center. Personalized learning should also be respected and integrated into the teaching of big data courses. Moreover, educational experts should be inspired to work collaboratively to improve the quality of big data MOOCs. It is essential to integrate cutting-edge disciplinary research, innovate course content, optimize the theoretical and practical teaching system, and create a multi-faceted knowledge acquisition model in the pre-situation introduction, in-class activity design, and post-class reflection. It is also crucial to pay attention to the collaboration among universities, businesses, and institutions, keep pace with the times in terms of development, and develop an advanced teaching philosophy. Moreover, in the overall planning of top-level designs, the implementation of curriculum teaching, the production of teaching videos, and in providing services, such as student Q&As, task release, and homework grading, a reasonable division of labor and rational personnel structures are important. The interaction between experienced expert-type teachers and energetic middle-aged and young teachers should also be emphasized, with their active participation in teaching reform to enhance the quality of education.

*4.3. Course Construction*

In China, the orientation of big data MOOCs is not clear. It is suggested that we use SMART standards to define teaching goals so that they can be measurable, achievable, realistic, and time-bound.

Particular emphasis should be placed on improving the proportion of intermediate-level and advanced-level courses, improving the high-level nature of big data courses, increasing the challenge level of courses, and meeting the comprehensive ability development requirements of students to solve complex big data problems. The principle objectives should be "student-centered, output-oriented, and continuous improvement".

Course designs should focus on interaction and feedback, and the diversification of teaching methods, such as video interactions, animation interactions, etc., which can be used to enhance students' immersive classroom experiences and learning effectiveness.

The use of information database management technology and a two-way interaction function is suggested. On the one hand, the MOOC platform can achieve a complete tracking record of each online student's personality data, learning process, stage, etc. Teachers and students can communicate with each other through the internet in an all-round way, narrowing the psychological distance between teachers and students, and increasing the opportunities and scope of communication between teachers and students. On the other hand, the teaching and learning service system can make personalized learning

recommendations for different students based on the personal data recorded by the MOOC platform. Through the statistical analysis of the types, numbers, and times regarding students' questions, teachers can understand the doubts, difficulties, and main problems encountered by students in learning, and guide students more pertinently. This is a realistic and effective way to realize individualized student-centered teaching.

On the basis of teacher evaluations and platform evaluations, it is suggested that teaching evaluations change to more learner-centered evaluation strategies and that peer evaluation and self-evaluation increase.

To incorporate character development into big data courses, elements of patriotism, technological advancement, and other core ideologies should be woven into the big data knowledge framework. This integration should achieve organic unity among values, knowledge, abilities, and qualities, creating a comprehensive educational classroom. Knowledge systems should be dynamically updated to provide students with a diverse range of learning resources in a timely manner, reflecting the ideological, scientific, and temporal aspects of the data science field. This integration will help students to effectively integrate and advance the development of their knowledge and skills, ultimately enhancing their overall satisfaction with learning.

## 5. Conclusions

To investigate the current situation of big data MOOCs in the intelligent era based on the perspective of improving the mental health of college students, and to summarize the construction experiences and areas for improvement, a three-dimensional MOOC analysis framework was developed, focusing on platform design, organizational mechanisms, and course construction. The construction of 525 big data courses on 16 MOOC platforms is compared and analyzed from three aspects: the platform (including platform construction, resource quantity, and resource quality), organizational mechanism (including the course opening unit, teacher team, and learning norms), and course construction (including course objectives, teaching design, course content, teaching organization, implementation, teaching management, and evaluation). Drawing upon the successful experiences of international big data MOOCs and excellent Chinese big data MOOCs, and considering the requirements of relevant documents, such as the no. 8 document (J.G. [2019]), no. 3 document (J.G. [2015]), no. 1 document (J.G. [2022]), as well as the "Educational Information Technology Standard CELTS-22—Online Course Evaluation Standard", recommendations about the platform, organizational mechanism, and course construction are provided for the future development of big data MOOCs in China.

To solve the problems in the construction and application of big data MOOCs in China, on the one hand, it is essential to offer suggestions for enhancing the quality of MOOCs, and bring experiences and enlightenment to MOOCs managers and teachers, to better promote the reform of education informatization based on MOOCs, and provide students with high-quality big data curriculum resources. On the other hand, it also provides help for the training of big data talents and the implementation and promotion of strategies.

In future research, the following two aspects can be explored in depth. Different theoretical bases can be selected as the starting points to study big data MOOCs from different perspectives, such as from the perspective of teachers and platform managers, to expand the research orientation of big data MOOCs horizontally and consolidate the specialization of big data MOOCs in China vertically. Moreover, user groups can be refined, based on different learning experiences, knowledge bases, and learning purposes. From the perspective of learners, an in-depth and detailed analysis of the current state of big data MOOCs' construction and application can be undertaken. This analysis will improve the construction mechanisms and application standards for big data MOOCs in China.

**Author Contributions:** Conceptualization, H.S., L.M. and N.M.; methodology, H.S., L.M. and N.M.; formal analysis, H.S., investigation, L.M.; data curation, H.S., L.M. and N.M.; writing—original draft preparation, H.S., L.M. and N.M.; writing—review and editing, H.S., L.M. and N.M.; resources, H.S., L.M. and N.M. Project administration, N.M. All authors have read and agreed to the published version of the manuscript.

**Funding:** This work was supported by the key education project "Research and Practice of Intelligent Interactive Technology in Intelligent Connected Vehicles" (grant no. 40041001202313) of the China Society of Vocational and Technical Education, and was supported by the 2022 "Industry-science Cooperative Education Project for "AI Profession intelligent Interactive Application Innovation Project Design and Practice" of the Ministry of Education, and was supported by the national level E-commerce First-class Undergraduate Major Construction Point of Beijing Union University

**Informed Consent Statement:** Not applicable.

**Data Availability Statement:** The dataset is available through the National Higher Education Smart Education Platform: https://www.smartedu.cn/. The corresponding author can make the data available upon request.

**Acknowledgments:** The authors would like to thank Zhuo Li, Li Yue, Zeyi Shan and Man Ye for their contributions to the data processing of this study.

**Conflicts of Interest:** The authors declare no conflict to interest.

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
