# Peer review of "Analysis of the Current Situation of Big Data MOOCs in the Intelligent Era Based on the Perspective of Improving the Mental Health of College Students"

_information, doi:10.3390/info14090511_

Round 1

Reviewer 1 Report

  1. Template: Missing authors' names and Affiliations.
  2. Clarity of Research Problem: The introduction effectively sets the broader context, however, the specific research question could be made more explicit. The statement of the research problem should be clear and concise to facilitate better understanding of the research objectives and methodology.

  3. Insufficient Context for Documents Referenced: The paper mentions the No. 1 Document (J.G. [2022]) and No. 8 Document (J.G. [2019]) from the Chinese Ministry of Education as significant foundations for the research, but does not adequately explain their relevance or content. To improve the paper, a brief, informative summary of these documents should be included, highlighting their relevance to MOOCs and big data education.

  4. Lack of Quality Assessment Framework: While the paper discusses course quality in terms of lecturer background and the existence of national quality courses, it lacks a comprehensive quality assessment framework for big data MOOCs. Such a framework should encompass various aspects of MOOCs like course content quality, pedagogical methods, student engagement and learning outcomes, and technology-enhanced features. This would allow for a more robust evaluation of the quality of the MOOCs under study.

  5. Inadequate Discussion on Industry-Academia Collaboration: The paper notes the lack of courses developed through industry-university-research cooperation, but fails to thoroughly explore the reasons for this, or propose strategies for enhancing such collaboration. The authors should elaborate on this, potentially incorporating survey or interview data from both industry and academia.

  6. Undefined Learning Objectives: The authors have identified that the learning objectives for big data MOOC courses lack clear definition. However, strategies to address this issue have not been proposed. The revised version could benefit from suggestions on setting SMART learning objectives or using a backward design approach.

  7. Insufficient Differentiation in Teaching Practices: The paper states that domestic big data MOOCs do not adequately cater to individual differences among students. Recommendations for addressing this issue, such as incorporating differentiated teaching practices or adaptive learning technologies, should be included in the revision.

  8. Limited Evaluation Methods: The paper discusses that current evaluation methods rely heavily on teacher evaluations and automated platform-based evaluations. The authors could suggest a shift towards more learner-centric evaluation strategies, such as peer assessments, portfolio assessments, and self-assessments.

  9. Ethical Concerns Regarding Data Privacy and Autonomy: The authors recommend strengthening the implementation of artificial intelligence and big data monitoring to monitor students’ learning processes, which could raise considerable ethical concerns. This strategy inherently involves collecting and analyzing a large amount of personal data from students, thereby introducing potential risks related to data privacy and autonomy. It is critical for the authors to address how students' privacy will be safeguarded, how their data will be utilized, who will have access to it, and for what purposes. Also, the potential misuse of this data and the safeguards that should be in place to prevent such misuse need to be discussed. If these concerns are not adequately addressed, there's a risk of infringing on student privacy and autonomy, which could lead to mistrust and discourage students from participating in MOOCs. Furthermore, this could present a significant barrier if these courses were to be marketed to international students or used in collaboration with foreign institutions, given the globally increasing focus on data protection and privacy rights.

In addition to these major points, it is recommended that the authors cite the following papers: doi.org/10.1177/01655515221116519 and doi.org/10.3390/bdcc7010047. These references could provide additional insights and support for the study's findings and recommendations.

The overall quality of the English language in the paper is satisfactory. However, there are some areas where improvement can be made to enhance clarity and readability. Some sentences are too long or complex, which can make them difficult to follow. In certain sections, the points being made could be clearer. The paper sometimes alternates between different terminology to describe the same concept. There are occasional grammatical and punctuation errors that should be corrected. 

Reviewer 2 Report

This paper examines the state of big data MOOCs and their impact on college students' mental health. By analyzing 525 courses on 16 platforms, the authors found issues such as unclear objectives and lack of personalization. They suggest improvements, such as AI monitoring of student progress, respecting individual learning styles, and introducing more interactive teaching methods. The following issues should be taken into consideration to further improve the paper quality.

1. I recommend a thorough revision of the introduction section. It should more explicitly focus on the problems currently faced in the field of travel information recommendation. This includes a review of the works done by some key researchers in this field, as well as a clear indication of the innovations introduced in your study. I suggest also adding an outline of the rest of the paper, which will help readers understand the flow and structure of your work.

2. I found that the literature review in introduction section needs to be enhanced. Instead of merely listing references, I encourage you to delve deeper into an analysis of the strengths and weaknesses of other studies. This approach will allow you to better highlight the novelty and contribution of your own work, and to situate it more clearly within the broader context of the existing literature. In particular, being more explicit about how your work extends or improves upon previous research would be beneficial.

3. The authors should introduce more related literatures in recent years to reflect the newly emerged research outcomes, such as the following ones: interaction-enhanced and time-aware graph convolutional network for successive point-of-interest recommendation in traveling enterprises; privacy-aware point-of-interest category recommendation in internet of things; an attention-based category-aware GRU model for the next POI recommendation.

4. The manuscript has several typographical errors that need to be corrected for consistency and clarity. Specifically, the term "MOOC" varies in capitalization throughout the text. Please revise the manuscript carefully to ensure consistent usage of the term, adhering to the correct form "MOOC", not "MOOc".

5. The manuscript currently lacks a distinct conclusion section. It is strongly recommended that you add a conclusion that encapsulates the innovative contributions of your research as well as the results achieved. The conclusion should succinctly summarize the main findings, highlight the novelty of your work, and discuss potential implications or future directions. 

Moderate editing of English language required.

Reviewer 3 Report

This manuscript mainly discusses the MOOC problem in the big data environment and proposes a three-dimensional MOOC analysis framework. It compares and analyzes the construction of 525 big data MOOC courses on 16 domestic and foreign MOOC platforms from three aspects: platform, organizational mechanism, and course construction. It points out the shortcomings and challenges in the construction of big data MOOC courses in China, and provides constructive improvement suggestions from three aspects. The three-dimensional analysis framework proposed in this paper has certain novelty, and its work has important reference value. 

The suggestions for this paper as follows: 

1) The paper only provides a simple quantitative comparison of the big data core course on MOOC, lacking further in-depth analysis. It is recommended to provide a clear evaluation standard and method for quantitative analysis, which is more convincing. It is recommended to supplement it. 

2) This paper provides relevant suggestions on the shortcomings and challenges faced in the construction of china MOOC courses, but these suggestions are relatively broad in content and face some problems in operability. I recommend to refine them.

Round 2

Reviewer 1 Report

The authors have successfully answered to all my questions.